# Accounting for Wood, Foliage Properties, and Laser Effective Footprint in Estimations of Leaf Area Density from Multiview-LiDAR Data

**François Pimont *, Maxime Soma and Jean-Luc Dupuy**

UR 629 Ecologies des Forêts Méditerranéennes (URFM), INRA, 84000 Avignon, France
* Correspondence: francois.pimont@inra.fr; Tel.: +33-432-722-947

**Abstract:** The spatial distribution of Leaf Area Density (LAD) in a tree canopy has fundamental functions in ecosystems. It can be measured through a variety of methods, including voxel-based methods applied to LiDAR point clouds. A theoretical study recently compared the numerical errors of these methods and showed that the bias-corrected Maximum Likelihood Estimator was the most efficient. However, it ignored (i) wood volumes, (ii) vegetation sub-grid clumping, (iii) the instrument effective footprint, and (iv) was limited to a single viewpoint. In practice, retrieving LAD is not straightforward, because vegetation is not randomly distributed in sub-grids, beams are divergent, and forestry plots are sampled from more than one viewpoint to mitigate occlusion. In the present article, we extend the previous formulation to (i) account for both wood volumes and hits, (ii) rigorously include correction terms for vegetation and instrument characteristics, and (iii) integrate multiview data. Two numerical experiments showed that the new approach entailed reduction of bias and errors, especially in the presence of wood volumes or when multiview data are available for poorly-explored volumes. In addition to its conciseness, completeness, and efficiency, this new formulation can be applied to multiview TLS—and also potentially to UAV LiDAR scanning—to reduce errors in LAD estimation.

**Keywords:** LAD; LAI; effective footprint; LiDAR; Maximum Likelihood Estimation; MLE; multiple view points; TLS; voxel; wood

## 1. Introduction

The amount and spatial distribution of foliage in a tree canopy have fundamental functions in ecosystems as they affect energy and mass fluxes through photosynthesis and transpiration [1]. Terrestrial Light Detection and Ranging (LiDAR), hereinafter referred to as Terrestrial Laser Scanning (TLS) recently emerged as a promising tool to estimate leaf or plant area density (LAD and PAD, in $m^{-1}$) distribution for individual plants and forest plots [2]. The approach can be applied to a variety of volumes of interest, assuming random distribution of vegetation inside. These volumes can be either horizontal layers to estimate LAD profiles [3–7], individual tree crowns [8], or voxels to estimate the tridimensional distribution of LAD [9–15]. In these different approaches, a traversal algorithm is applied to each volume of interest to compute gap fractions, hits, and for some approaches, "free paths" (i.e., distances travelled without interception, in m), in order to derive different metrics to estimate the quantity of interest [3–15].

Among the different metrics suggested in the past, a recent comprehensive theoretical study [15] has shown that the Modified Contact Frequency, first introduced in a previous study [9], corresponds to the Maximum Likelihood Estimator (MLE) [16] of the attenuation coefficient. This attenuation coefficient is the rate at which the point cloud density decays with vegetation interception, which is

related to the LAD and PAD linearly. This attenuation coefficient, however, is more often estimated by inverting the equation of the transmittance, which decays exponentially with the attenuation coefficient. This approach is referred to as the Beer's law-based (or gap fraction) method. To date, Beer's law-based methods are still more popular than the MLE [2], although they do not take full advantage of the tridimensional information available in the point cloud, by ignoring free paths, which leads to additional complexity in the inversion when the path length is not constant (simple cosine term in gap fraction methods, but complex corrections in crown volumes [8] and voxels [15,17]). This trend can probably be explained by the strong legacy of gap fraction approaches in this research field, which has been focused on 2D sensors, such as hemispherical photographs or Licor LAI-2000 (a popular Plant Canopy Analyzer based on multiple light intensity measurements at different zenith angles), prior to the emergence of more expensive and more complex 3D sensors. The benefits of the MLE are that the formulation is more straightforward and efficient, without making any assumption on the geometry of the volume of interest [15]. The method provides the most likely estimate of the attenuation coefficient given the observation of free paths and hits, simply assuming that explored and unexplored regions exhibit similar random distributions of vegetation elements. The MLE approach, which relies on free paths, should not be confused with the PATH method [6,8], which uses the path-length distribution to identify crown volumes in order to mitigate the impact of clumping in crown volumes, and which has to date only been applied to Beer's law-based methods. One could notice that the PATH method could be combined with MLE instead. One limitation of the MLE (but also of Beer's law-based methods) is their bias when the number of beams exploring a given voxel is limited (typically smaller than 30), or when vegetation elements are not small with respect to voxel size. Such biases can be theoretically corrected, leading to a bias-corrected MLE that is "efficient" in the sense that it is unbiased and it exhibits the smallest variability theoretically reached by any unbiased estimator [15].

The estimator presented in a previous study [15], however, is based on simplifying theoretical assumptions—vegetation elements are assumed to be randomly distributed within volumes and TLS beams are infinitely thin. Hence, it typically requires additional corrections when applied to actual point clouds to account for LiDAR effective footprint in clumped vegetation elements [14], similar to other methods applied to voxels or tree crowns [8–14]. Also, the theoretical formulation presented in a previous study [15] neglects the presence of woody elements in the estimation of LAD, which should be accounted for separately, either using a separation between leaf and wood returns [9,18] or "leaf-off" scans [8,14]. To date, a theoretical framework for such inclusion is still missing. Another limitation of the theoretical formulation is that it was applied to an individual scan, whereas field applications often require the use of multiple viewpoints to mitigate the impact of vegetation occlusion. Several methods have been suggested to combine the information arising from the different scans, such as relying on the best viewpoint on a given voxel (i.e., the one with maximal beam number [19]), combining all beams as if they belonged to the same scan [9], or weighting estimates from each scan according to the number of beams of each viewpoint [8,11]. To date, the consequences of such combinations on LAD estimation have never been studied.

In the article, we present a bias-corrected Maximum Likelihood Estimator for the LAD with multiview-LiDAR data in volumes of interest, which naturally extends the formulation presented in a previous study [15] to actual field data, with the presence of wood volumes, wood hits, correction terms to account for beam divergence, and vegetation clumping, as well as to multiview data. The new method is then briefly compared with former approaches in two simple numerical experiments.

## 2. Background and Limitations of Existing Methods

### 2.1. The Theoretically Bias-Corrected Estimator (TBC-MLE)

Here, we briefly summarize the *PAD* estimation in the mathematical framework proposed in a previous study [8]. This approach is based on the following equation:

$$\widehat{PAD} = \frac{H}{G}\widetilde{\Lambda}, \tag{1}$$

where $\widetilde{\Lambda}$ (in $m^{-1}$) is an estimator of the attenuation coefficient, $G$ is the dimensionless leaf projection factor, and $H$ is a dimensionless correction factor that accounts for the laser effective footprint in clumped vegetation [14]. Observations suggest that $H$ decreases with the distance to the scanner to compensate for the increase in effective footprint caused by beam divergence and variation in return detection, which induces an increase of the apparent area of vegetation elements [14,18]. Also, $H$ increases with the voxel size to compensate for the effect of vegetation clumping inside voxels, which causes discrepancies to the theoretically random distribution of vegetation elements, as a consequence of Jensen's convexity inequality [12,14,18,20]. It also depends on the scanner, and to a lesser extent, on foliage morphological differences between species [14], although the element size and shape can at least partially be accounted for through the notion of "effective" free path $z_e$ (in m, see a previous study [15] or Equation (3) below and Appendix A). The dimensionless projection function $G$ can be separately estimated [9,21].

For a given viewpoint, the attenuation coefficient can be estimated from the Maximum Likelihood estimator (MLE). It is equal to the number of hits $Ni$ divided by the sum of free paths $\Sigma z$, in m (Figure 1), which are computed with a traversal algorithm:

$$\widetilde{\lambda} = \frac{Ni}{\Sigma z} \tag{2}$$

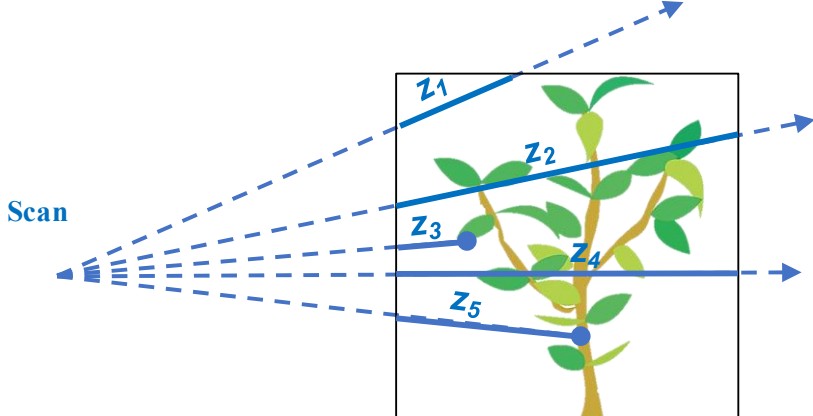

**Figure 1.** Scheme of the information provided by the traversal algorithm which is used to compute the MLE of the attenuation coefficient: number of hits $Ni$ (blue dots) and free paths (distances $z$ travelled by the beams; blue lines) in each voxel. The dotted lines represent pulse trajectory.

The free path sum is the total distance actually travelled by beams inside a voxel before their eventual interceptions by a vegetation element, which can be either leaf or wood (Figure 1). This approach differs from the Beer's law-based method, which does not use the information provided by free path lengths. Indeed, it estimates the empirical transmittance (as $1 - \frac{Ni}{N}$, where N is the number of beams entering the voxel) before inverting the transmittance equation, which can be complex when the path length (intersection between beam trajectory and voxel) is variable [15].

This MLE estimator (Equation (2)) is similar to the Modified Contact Frequency introduced in a previous study [9]. This estimator is biased when the beam number N is low or when vegetation elements are not infinitely small, and it can be corrected with a more sophisticated estimator $\widetilde{\Lambda}$, referred to as the theoretically bias-corrected MLE (TBC-MLE [14,15]). In this estimator, each free path $z$ is replaced by the effective free path $z_e$ (in m):

$$z_e = -\frac{log(1 - \lambda_1 z)}{\lambda_1},$$

(3)

where $\lambda_1$ is the attenuation coefficient, in m$^{-1}$, of a single element of vegetation (see Appendix A for an estimation of $\lambda_1$ for cylindrical needles or elliptical flat leaves). Obviously, $z_e \approx z$ when $\lambda_1$ is very small (i.e., the turbid medium assumption).

For the purpose of the present study, the TBC-MLE of the PAD [14] is slightly rearranged to ease generalization of multiple viewpoints, which is proposed in the next section:

$$\widetilde{PAD} = \frac{H}{G}\widetilde{\Lambda} = \frac{H}{G\sum z_e}\left(Ni - \frac{\sum_{hits} z_e}{\sum z_e}\right)$$

(4)

In Equation (4), $Ni$ is the number of hits in the voxel, whereas $\Sigma z_e$ is the effective free path sum, and $\Sigma_{hits} z_e$ is the effective free path sum for beams with hits inside the voxel (hence $\frac{\sum_{hits} z_e}{\Sigma z_e}$ ranges between 0 and 1). The second term in brackets corresponds to the bias-correction term suggested in a previous study [15], which can be neglected when the beam number is high (i.e., larger than 30). This estimator is unbiased in a wide range of vegetation element size and density when $N > 5$ and reaches the Cramer-Rao bound, meaning it is the most efficient unbiased estimator given the available information [15].

In this formulation, $H$Ni is close to the number of hits centered on a leaf, first introduced in a previous study [9], to account for beam divergence in the modified contact frequency formulation. Our formulation, however, is slightly different from [9], since ignored beams with partial hits in their "volume fraction" are summed (see Equation (12) in the previous study [9]), which would be equivalent to ignoring beams with partial hits in the free path sum $\sum z_e$ present at the denominator of Equation (4) above.

In Section 3, we rigorously account for $H$ and $G$ in mathematical derivations.

*2.2. Theoretical Variance and 68% Confidence Interval of the TBC-MLE*

Mathematical derivations presented in a previous study [15] led to an estimator of the variance of $\widetilde{PAD}$:

$$\sigma^2_{\widetilde{PAD}} = \left(\frac{H}{G}\right)^2 \sigma^2_{\widetilde{\Lambda}} = \frac{1}{Ni}\left(\frac{1}{\frac{G}{H}\Sigma z_e}\left(Ni - \frac{\sum_{hits} z_e}{\sum z_e}\right)\right)^2$$

(5)

Such a variance estimator is useful to quantify the accuracy of a given *LAD* estimate, since the variance measures the magnitude of estimation errors. In Equation (5), we only accounted for the random variations associated with LiDAR sampling in the voxel, which mostly depend on the total number of beams entering the voxel [15]. For simplicity, we neglected a second term, which arises from the variability of element positions in the vegetation sample present in the voxel. According to the previous study [15], this quantity significantly contributes to the overall error when vegetation elements are not numerous and when beam number is low. For the interested reader, an empirical model for this quantity was presented in the previous study [15], in the case of "square flat" leaves. A related metric of interest is the radius of the 68% confidence interval of the LAD estimate, which is given by [15]:

$$\Delta\widetilde{PAD} = \frac{H}{G}\Delta\widetilde{\Lambda} = \frac{1}{G/H}\frac{Ni + \frac{1}{2} - \frac{\sum_{hits} z_e}{\Sigma z_e}}{\sqrt{Ni + \frac{1}{2}\Sigma z_e\left(1 + \frac{1}{N}\right)}},$$

(6)

where $N$ is the total number of beams entering the voxel.

The rationale for the $\frac{1}{2}$ terms is to avoid the confidence interval radius from equaling 0 when Ni = 0, which would clearly be incorrect. Indeed, zero hit in a voxel does not necessarily mean that no vegetation elements are present, but only indicates that current sampling beams have not detected any vegetation element. In other words, there is a non-zero chance that vegetation elements are present. This confidence interval is referred to as "Agresti-Coull" in the previous study [15] and leads to a lower bound of $\frac{1}{\sqrt{2}\Sigma z_e\left(1+\frac{1}{N}\right)}$ when Ni = 0. It expresses that the estimation is more accurate as $\Sigma z_e$ increases, but never reaches 0, even for a high number of beams $N$.

### 2.3. Accounting for Wood Returns

As most applications focus on LAD and not PAD, several methods have been developed to account for wood elements. From a separation of leaf and wood returns based on return intensity, the authors of a previous study [9] suggested that beams corresponding to wood hits be ignored in their formulation of the modified contact frequency, leading to the following (simplified) estimator:

$$\widetilde{LAD} = \frac{\text{Ni}^l}{G\sum_{\neq wood\ hits} z},\tag{7}$$

where $\text{Ni}^l$ is the number of leaf hits and $\sum_{\neq wood\ hits} z = \Sigma z - \sum_{wood\ hits} z$ corresponds to the sum of free paths for beams that do not correspond to a wood hit.

A similar idea was also applied to Beer's law-based method [17], leading to

$$\widetilde{LAD} = -\frac{\log\left(1 - \frac{\text{Ni}^l}{\text{N}^{\neq w}}\right)}{\delta},\tag{8}$$

where $\text{N}^{\neq w} = N - \text{Ni}^{wood\ hits}$ is the total number of beams in the volume of interest that do not correspond to wood hits, and $\delta$ is the path length (in m, assumed constant for simplicity).

Another approach was to determine the *LAD* as a difference between "leaf on" and "leaf off" conditions [8,14]. This approach relies on the implicit assumption that the total attenuation coefficient of vegetation elements is the sum of the attenuation coefficients of leaf and wood elements, respectively, which requires an assumption of random distribution for both leaf and wood elements, which is obviously incorrect in the case of logs or large branches. This is equivalent to the introduction of a multiplicative factor equal to the leaf hit fraction $F$:

$$\widetilde{LAD} = F\widetilde{PAD}, \text{ with } F = \frac{\text{Ni}^l}{\text{Ni}}\tag{9}$$

This approach can be applied to either the Beer's law-based method or the MLE method, but the resulting estimators differ from Equations (7) and (8) above, in which beams with wood hits are ignored. To date, these methods have never been compared. Finally, in these three approaches, the volume occupied by logs and branches inside the voxel was neglected. In Section 3, we rigorously include wood volumes and leaf hits in the mathematical derivations.

### 2.4. Multiview Estimators

When several points clouds are available (each with an index $j \in [1; J]$), the most basic method to deal with multiview data is to select the "best viewpoint" (i.e., the scan $j$, which sampled a given voxel with the highest number of beams $N_j$), as in a previous study [19]. This estimator, shown here for an *LAD* estimator, referred to as "*Nmax*", is defined as:

$$\widetilde{LAD}^{Nmax} = \widetilde{LAD}_{jmax}, \text{ with } jmax \text{ so that } N_{jmax} = \max_{j \leq J} N_j\tag{10}$$

This approach is unbiased, provided that each individual estimator is unbiased (e.g., when $N > 5$ with the TBC-MLE [15]). However, information from other scans is ignored, which is not optimal, especially when several viewpoints explore a given voxel with similar numbers of beams.

A more sophisticated method, referred to as "*N-weighted*" (NW), is based on a weighted average of each estimate $\widetilde{LAD}_j$ (from the different viewpoints), with the weights being equal to $N_j$, as suggested in previous studies [8,11]:

$$\widetilde{LAD}^{NW} = \frac{1}{\sum_{j \leq J} N_j} \sum_{j \leq J} N_j \widetilde{LAD}_j \tag{11}$$

No information is ignored with this second approach, since all viewpoints contribute to the final estimation.

## 3. Generalized Maximum-Likelihood Estimation for LAD from Multiview-LiDAR Data

This section details our new formulation of the estimation of Leaf Area Density from multiview LiDAR data within a volume of interest, which can be either a voxel or a crown volume, but it is simply referred to as "the voxel" for simplicity. It relies on similar assumptions as above, with three noticeable differences. First, we explicitly consider the sub-volume $V_w$ (in m$^3$) of the voxel $V$ (in m$^3$) occupied by wood elements (Figure 2). Within a voxel volume $V$, we assume that small leaf elements are randomly distributed in the sub-volume $V - V_w$ of $V$, which is not occupied by the wood. This sub-volume containing the leaf elements has a (dimensionless) volume fraction $\alpha$ equal to:

$$\alpha = 1 - \frac{V_w}{V} \tag{12}$$

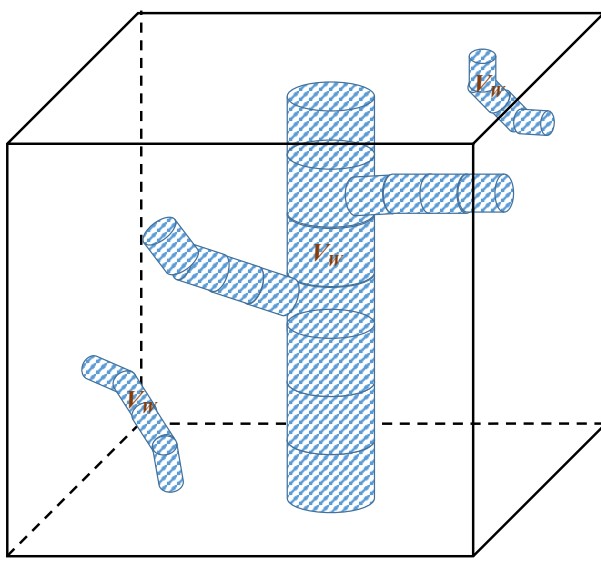

**Figure 2.** Scheme of the representation of wood volumes $V_w$ (in dashed blue) in the voxel of volume $V$. We assume that leaf elements are randomly distributed in volume $V - V_w$, which exhibits a very complex and unknown topology.

In general, $\alpha$ is very close to 1, except when large branches or logs intersect the voxel. Here, no specific assumption is made on the topology of the wood volume $V_w$, neither on how it is distributed with respect to the volume $V - V_w$, in which leaves were present. In practice, $\alpha$ can be estimated from the intersection between the voxel and tree models, which can be derived from LiDAR data [22].

Second, we assume that the effective attenuation coefficient in $V - V_w$, which corresponds to what is actually viewed by the scanner from viewpoint j, verifies $\lambda_j = \frac{G_j LAD}{H_j}$ and that the factors for effective

footprint on clumped vegetation $H_j$ and for leaf projection $G_j$ are known (using methods described in previous studies [14,21,23], for example). In this framework, the $\lambda_j$ value defines the probability distribution function of any laser beam entering the voxel of interest (see Appendix B, Equation B1, for details) and no multiple echoes exist. Third, we assume that $J$ point clouds are available (each with an index $j \in [1; J]$). It is important to acknowledge that correction factors can exhibit large variations with scanner position $j$ for a given voxel, as distances to scanner or view angle differ. In Appendix B, we apply similar mathematics as in the previous study [15] to leaf elements distributed inside $V - V_w$. For consistency with usual definitions, the *LAD* is still defined as the surface area of leaf elements divided by the voxel volume $V$, even though the leaves are not distributed in the whole volume $V$. This explains the presence of volume fraction $\alpha$ in the following equations. From the distribution of "multiview" leaf hits, free paths, projection factors, and correction factors, the objective here is to determine the most likely value of *LAD* (MLE, given the observations. The mathematical derivations slightly differ from the previous study [8], since there is not a single attenuation coefficient $\lambda$ for which the MLE can be computed, but there are as many attenuation coefficients $\lambda_j$ as viewpoints $j$. Thus, we directly compute the Maximum Likelihood Estimator "MLE" of the *LAD* in m$^{-1}$ (i.e., not of the attenuation coefficient $\lambda$), which cancels the first derivative of log-likelihood [16] of the LAD and find (Equation (B6)):

$$MLE_{LAD}^{M} = \alpha \frac{Ni^l}{\sum \frac{G}{H} z_e}, \tag{13}$$

where $Ni^l = \sum_j Ni_j^l$ is the total number of leaf hits (for all scans) and $\sum \frac{G}{H} z_e = \sum_{j=1}^{J} \sum_{i=1}^{N_j} \frac{G_j}{H_j} z e_j^i$ is the sum of the products $\frac{G_j}{H_j} z_j^i$ for beams exploring $V - V_w$ (Figure 3). The "M" superscript corresponds to "Multiview". Here, it is important to note that according to the mathematics, wood hits are ignored in the count of hits, but not in the free-path sum, contrary to what was suggested in a previous study [9]. Also, the correction factor $\frac{G}{H}$, which accounts for differences between viewpoints, appears as a multiplicative factor in the free path sum. Hence, all hits should be considered equally in the hit sum, no matter the distance to the scanner or the view angle, but the free paths should be modified to account for these differences. As for the wood hits, this slightly differs from the "center leaf hit" method presented in previous studies [9,18].

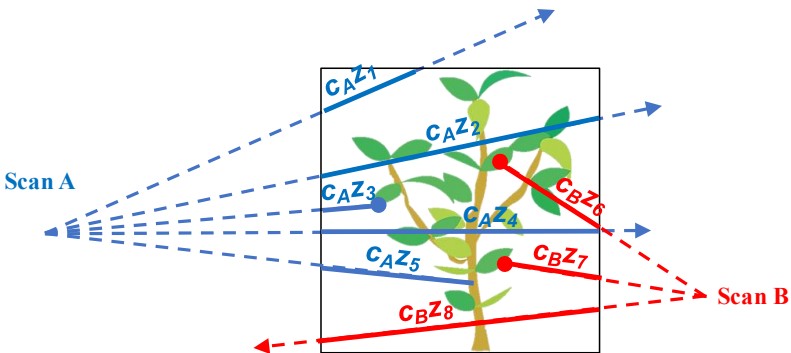

**Figure 3.** Scheme of the information provided by the traversal algorithm, which is used to compute the Maximum Likelihood Estimator (MLE) of the Leaf Area Density (*LAD*) from multiview data from Scan A (in red) and Scan B (in blue): leaf hits (blue and red dots) and free paths (distances $z$ travelled by the beams; blue and red lines) in the voxel. The dotted lines represent pulse trajectories; $c_A = \frac{G_A}{H_A}$ and $c_B = \frac{G_B}{H_B}$ represent the correcting factors for viewpoints A and B, respectively, which differs with distance to scanner and view angle. For simplicity, correction for effective free path ($z_e$; Equation (3)) is ignored. Note that in this framework, no leaf can be distributed within the volume $V_W$ occupied by wood elements (in brown). Also, and contrary to Figure 1, the hits corresponding to woody elements (e.g., 5th beam of scan 1) are ignored in the hit sum, but the corresponding free paths are accounted for in the free-path sum, in which $c_A$ and $c_B$ are used as multiplicative factors.

As for a single viewpoint, this "MLE" is biased when the number of beams is low and a correction can be computed [15]. Generalizing this correction to the multiview LAD estimator ("$M$") led to (Appendix B):

$$\widetilde{LAD}^M = \frac{\alpha}{\sum \frac{G}{H} z_e} \left( Ni^l - \frac{\sum_l \frac{G}{H} z_e}{\sum \frac{G}{H} z_e} \right), \tag{14}$$

with $\sum_l \frac{G}{H} z_e$ corresponding to the sum of $\frac{G_j}{H_j} z^i_j$ for beams corresponding to leaf hits only. This formulation obviously generalized the single-scan estimator $\widetilde{LAD}_j$, as re-wrote in Equation (4).

In practice, however, the formulation of Equation (14) requires discrimination of each hit depending on whether it is foliage or wood in order to compute the bias correction term. A slightly more practical formulation can be achieved assuming that $\sum_l \frac{G}{H} z_e \approx F \sum_{hits} \frac{G}{H} z_e$, with the hit leaf fraction $F = \frac{Ni^l}{Ni}$:

$$\widetilde{LAD}^M = \frac{\alpha F}{\sum \frac{G}{H} z_e} \left( Ni - \frac{\sum_{hits} \frac{G}{H} z_e}{\sum \frac{G}{H} z_e} \right) \tag{15}$$

Similarly, generalizing Equations (5) and (6), the variance of $\widetilde{LAD}^M$ is:

$$\sigma^2_M = \frac{\alpha^2}{Ni^l \left( \sum \frac{G}{H} z_e \right)^2} \left( Ni^l - \frac{\sum_l \frac{G}{H} z_e}{\sum \frac{G}{H} z_e} \right)^2 \approx \frac{\alpha^2 F}{Ni \left( \sum \frac{G}{H} z_e \right)^2} \left( Ni - \frac{\sum_{hits} \frac{G}{H} z_e}{\sum \frac{G}{H} z_e} \right)^2 \tag{16}$$

and the radius of the 68%-level confidence interval of LAD estimate is:

$$\Delta \widetilde{LAD}^M = \alpha \frac{Ni^l + \frac{1}{2} - \frac{\sum_l \frac{G}{H} z_e}{\sum \frac{G}{H} z_e}}{\sqrt{Ni^l + \frac{1}{2} \sum \frac{G}{H} z_e \left( 1 + \frac{1}{N} \right)}} \approx \alpha \frac{F \left( Ni - \frac{\sum_{hits} \frac{G}{H} z_e}{\sum \frac{G}{H} z_e} \right) + \frac{1}{2}}{\sqrt{FNi + \frac{1}{2} \sum \frac{G}{H} z_e \left( 1 + \frac{1}{N} \right)}} \tag{17}$$

The value of $F$ can be determined from one of the algorithms and methods developed to discriminate leaf and wood returns [18,24–29].

## 4. Numerical Experiments

Several aspects of the formulation presented in Section 3 have already been evaluated in a previous study [15]. We can cite the bias corrections for finite elements with the notion of effective free path $z_e$ (Equation (3)) and for small beam numbers (Equation (4)), as well as the efficiency of the MLE approach for random error minimization and the estimation of variance and confidence intervals. Here, we present two numerical experiments that aim at demonstrating the added value of the generalized formulation presented in Section 3, of which the results are compared to results from earlier formulations (i) to account for wood volumes and returns (Section 4.1), and (ii) to include multiview point clouds in (Section 4.2). In order to focus each experiment on the aspect of interest, we assumed for simplicity that vegetation elements are infinitely small, which simplifies the representation of vegetation and point cloud simulations, as described in a previous study [15].

### 4.1. Comparison between Formulations to Account for Wood Returns and Volumes

Experiment description

The experiment was carried out at the voxel scale, as in a previous study [15]. A cubic voxel of 0.2 m width was crossed by a vertical cylindrical branch of 0.05 cm radius, centered in the voxel. The cylinder is surrounded by randomly distributed and oriented infinitely small vegetation elements of constant LAD and the voxel is scanned by 500 horizontal LiDAR beams, which can be simulated using Equation C6, which was implemented in MATLAB scripts [15,23]. Here, beam interceptions by the branch were considered so that wood hits occurred. We assumed that LiDAR beams were infinitely

small so that the *H* correction factor was equal to 1. We repeated the experiment with 200 LAD values, randomly chosen between 0 and 4 m$^{-1}$. In this simple context, the volume fraction $\alpha$:

$$\alpha = 1 - \frac{\pi 0.05^2 0.2}{0.2^3} \simeq 0.804, \tag{18}$$

which means that 20% of the voxel was occupied by woody elements.

　　　The leaf fraction F corresponding to the different simulations were plotted as a function of reference LAD values in Figure A4. This ranged between 0 for very low LAD values to 0.4. This means that wood returns represent the majority of hits in all this experiments. This specific design (majority of wood hits, 20% of the volume occupied by wood) is not representative of most canopy volumes, but was chosen to emphasize differences between formulations.

　　　We then computed the estimations for six different leaf and wood formulations (Table 1). The first three formulations neglected the wood volume. The first one corresponds to the formulation of the modified contact frequency with wood hits (Equation (7)), as suggested in a previous study [9]. The second corresponds to the Beer's law-based formulation (Equation (8)), as suggested in another study [17]. The third formulation corresponds to Equation (15), but the equation was simplified. First, we used free path z (instead of effective free paths $z_e$, since element size is negligible); second, we could neglect the bias correction term due to low beam numbers (since $\text{Ni}^l \gg 1$); third, for a fair comparison with Equations (7) and (8), we temporary neglected the role of the wood volume, simply assuming that $\alpha = 1$. The other three estimators were the same, but the true value of $\alpha$ was incorporated as a multiplicative factor. Hence, the last estimator corresponds to Equation (15) (beyond the simplifications detailed below).

**Table 1.** Different estimators used to for numerical experiment described in Section 4.1.

| Equation | Simplified for Mulation | Reference |
|:---:|:---:|:---:|
| Equation (7) | $(a)\,\dfrac{\text{Ni}^l}{G\sum_{\neq wood\ hits} z}$ | [9] |
| Equation (8) | $(b)\,-\dfrac{\log\left(1-\frac{\text{Ni}^l}{N\neq w}\right)}{\delta}$ | [17] |
| Equation (15) (with $\alpha = 1$, $\text{Ni}^l \gg 1$ and $\lambda_1 \ll 1$) | $(c)\,\dfrac{\text{Ni}^l}{G\Sigma z}$ | This publication |
| Equation (7), with $\alpha$ multiplicative factor | $(d)\,\dfrac{\alpha \text{Ni}^l}{G\sum_{\neq wood\ hits} z}$ | [9] and this publication |
| Equation (8), with $\alpha$ multiplicative factor | $(e)\,-\dfrac{\alpha \log\left(1-\frac{\text{Ni}^l}{N\neq w}\right)}{\delta}$ | [17] and this publication |
| Equation (15) ($\text{Ni}^l \gg 1$ and $\lambda_1 \ll 1$) | $(f)\,\dfrac{\alpha \text{Ni}^l}{G\Sigma z}$ | This publication |

Results

　　　Figure 4 shows the comparison between predicted and reference LAD values for 200 simulations. All formulations led to an overestimation, with mean biases ranging between 24% and 64%, with the exception of Equation (15), which was unbiased (Figure 4f). The spread of the simulations around the fitted linear trend (dashed blue line) occurred because of the number of beams used in the present simulation (500) and would be much smaller with a higher number of beams, with lower RMSE (expressed in % of the mean reference LAD). The difference between the dotted line and the 1-1 line (in black) shows the potential biases of the different estimators, which were quantified by the mean bias (expressed in % of the mean reference LAD).

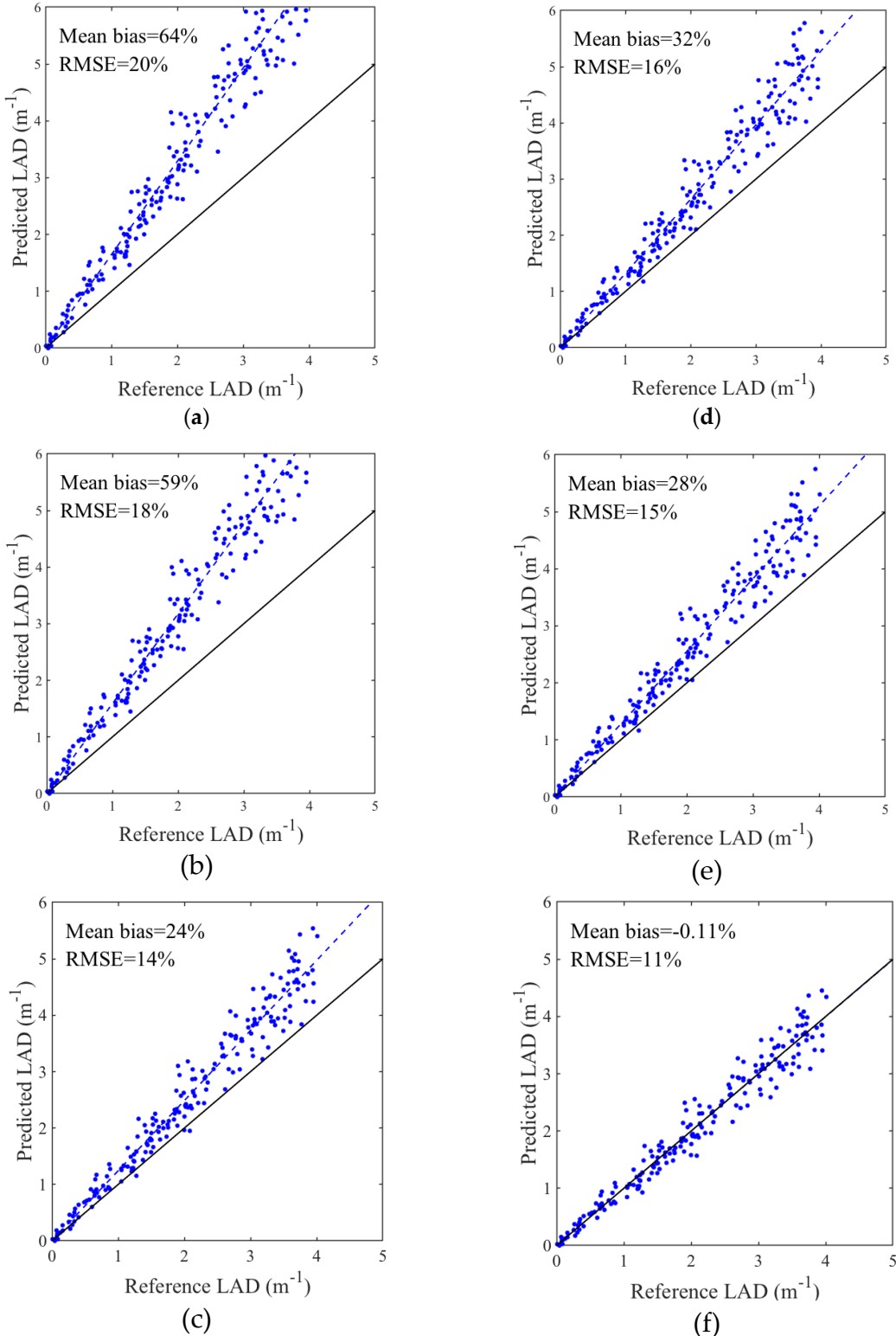

**Figure 4.** Comparison between predicted and reference LAD for a variety of formulations to account for wood in estimators (see Table 1 for details): (**a**) Equation (7); (**b**) Equation (8); (**c**) Equation (15), (with $\alpha = 1$, $Ni^l \gg 1$, and $\lambda_1 \ll 1$); (**d**) Equation (7), with $\alpha$ multiplicative factor; (**e**) Equation (8), with $\alpha$ multiplicative factor; (**f**) Equation (15) ($Ni^l \gg 1$ and $\lambda_1 \ll 1$). Formulations presented in subplots (**a**–**c**) ignored wood volumes, contrary to subplots (**d**–**f**).

Comparing subplots (a), (b), and (c) with (d), (e), and (f), respectively, demonstrated the important improvement associated with the volume fraction factor $\alpha$, which was especially important in the context where wood volume occupied around 20% of the volume of interest. The bias would obviously decrease if the wood volume was smaller, but this example shows that this factor should not be neglected in some cases (near logs and trunks in particular). Subplots (d) and (e) show that ignoring beams with wood hits in estimators was incorrect. Another limitation of these last two estimators is that their biases vary with the location of wood volumes inside the voxel, contrary to Equation (15), which is insensitive to wood volume distributions (provided that leaves are randomly distributed outside these volumes, as in our simulation). For example, the mean bias presented in Figure 4d reached 53% when the branch was located near the trailing face of the voxel (where beams leave the voxel), whereas it was limited to 8% when the branch was located near the leading face of the voxel (where beams enter the voxel), instead of 32% when the branch was centered (as in Figure 4d).

### 4.2. Comparison between Multiview Formulations

Experiment description

This second numerical experiment was carried out at the scale of a small forestry plot. The $\widetilde{LAD}^{M}$ differed from the "Nmax" multiview combination of $\widetilde{LAD}_j$ (Equation (10)), but also from the *"N-weighted"*, which can be shown with a numerical expansion of Equation (11). Beyond the conciseness and the mathematical support for Equation (15), it was important to quantify the error reduction resulting from the new formulation in "field-like" conditions. Thus, we conducted a numerical experiment corresponding to plausible field features, aiming at (i) providing a brief validation of the "M" multiview estimator of LAD presented above (Equation (15)), and (ii) comparing its performance with the two usual formulations to combine single-view estimates.

All of the details regarding this numerical experiment were provided in Appendix C for conciseness. In brief, we generated a "reference" $LAD_{ref}$ in a 10-m tri-dimensional mesh grid corresponding to plausible features in terms of LAI, clump size and vertical distribution [23]. Voxel size was equal to 0.1 m, and the cubic vegetation scene had a 10-m lateral extension (and a 10-m height). $LAD_{ref}$ corresponded to a clumped spatial distribution simulated from *RandomFields* R package, which was parameterized to correspond to realistic features of natural vegetation (cover fraction of 70% and LAI of about 3.8). The mean clump size, which was representative of the tree crown diameter, was 4 m. Additional clumping (~1 m) occurred to represent branch scale heterogeneity. The *LAD* vertical profile exhibited a peak around 7 m in height (Figure A1a).

We simulated five point clouds from different viewpoints. We then estimated the LAD using the three multiview formulations after applying a traversal algorithm to each point cloud to compute the different statistics. In this experiment, we assumed infinitely small vegetation elements, randomly distributed inside 10 cm voxels, so that no clumping occurred below 10 cm. We also neglected the wood volume (already investigated in Section 4.1) in order to focus on differences arising from multi-scan formulations.

Results

The mean biases observed in voxels, computed for three classes of beam number $N$, are shown in Table 2. With the new multiview estimator ($\widetilde{LAD}^{M}$), biases were smaller than 1% for $N \geq 10$ and were only equal to 2.2% when $N < 10$. The two other estimators exhibited biases of larger magnitudes, especially the "*N-weighted*" estimates ($NW$), which reached $-15\%$ when $N < 10$. Such a result was quite surprising; as a weighted average of unbiased estimators (computed for each scan), one would have expected the $NW$ estimator to be unbiased too. There was a simple explanation to this apparent paradox. When $N$ was smaller than 10, it often corresponded to cases where the beam number exploring a voxel from one or several viewpoints was smaller than 5, and in particular equal to 1 or 2. In these cases, the

single-view estimator was negatively biased [15]. For example, this bias was especially obvious when $Nj=1$ (in this case, it is equal to 0 when $Ni_j^l = 0$, but also when $Ni_j^l = 1$, since $\frac{\sum_l z_{e,j}}{\sum z_{e,j}} = 1$; see Equation (4)).

**Table 2.** Mean biases (in % of the mean $LAD_{ref}$) of the three estimators for three different classes of total beam number $N$.

| Range of Beam Number | $\widetilde{LAD}^{Nmax}$ | $\widetilde{LAD}^{NW}$ | $\widetilde{LAD}^{M}$ |
|:---:|:---:|:---:|:---:|
| N ≥ 2 and N < 10 | −6.0% | −15% | +2.2% |
| N ≥ 10 and N < 15 | +0.8% | −2.8% | +0.4% |
| N ≥ 15 | +0.0% | −0.4% | +0.0% |

Hence, the new multiview estimator ($\widetilde{LAD}^{M}$) was only marginally biased in all conditions, contrary to the other formulations. This situation was, in practice, quite frequent for voxels in which the total beam number was smaller than 10, as show in Figure 5, which represents the profiles of frequencies of four beam number classes in the virtual forest plot.

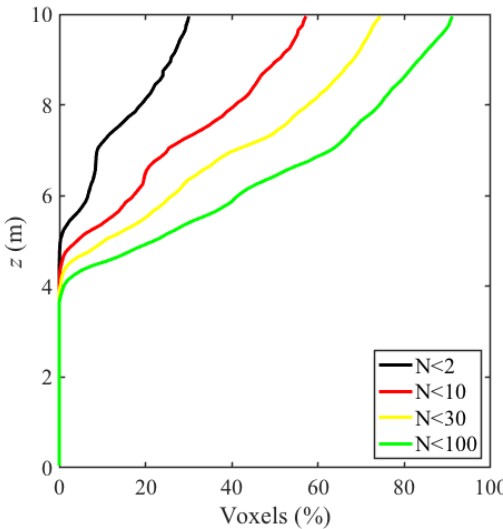

**Figure 5.** Vertical profiles of percentages of voxels with number of beams smaller than 2, 10, 30, and 100, in the numerical experiment described in Appendix C (five different viewpoints located at 1 m above the ground).

The RMSE observed in voxels, computed for six classes of beam number $N$, are shown in Table 3. With the multiview estimator ($\widetilde{LAD}^{M}$), RMSE were smaller than those of the two other estimates. In particular, differences between $\widetilde{LAD}^{M}$ and $\widetilde{LAD}^{Nmax}$ were observed for all classes of beam numbers and were explained by the fact that the information from secondary viewpoints was ignored with "*Nmax*", leading to larger RMSE. Differences between $\widetilde{LAD}^{M}$ and $\widetilde{LAD}^{NW}$ mostly occurred for $N$ ranging between 10 and 30, but RMSE for $\widetilde{LAD}^{NW}$ could be more than twice as big as for $\widetilde{LAD}^{M}$. More detailed analyses (not shown) showed that these strong differences in performances were caused by a very limited number of voxels, in which errors of $\widetilde{LAD}^{NW}$ were very high when compared to those of $\widetilde{LAD}^{M}$. This occurred when one of the $\widetilde{LAD}_j$ estimates with a very low number of beams ($Nj$ lower than 5) was very far beyond the reference value (for example, when the mean free path from viewpoint $j$ was unluckily very small for the $Nj$ beams). In this configuration, very large overestimations could occur for the "*N-weighted*" estimator, despite the weighting procedure, which was not able to dampen such outliers. As a result, the "*NW*" estimator led to higher RMSE than the "*Nmax*", despite more

information being used. Such differences were caused by infrequent but very large overestimations observed with $\widetilde{LAD}^{NW}$.

**Table 3.** Root Mean Square Error (in % of the mean *LAD*) of the three multiview estimators for six different classes of total beam numbers.

| Range of Beam Number | $\widetilde{LAD}^{Nmax}$ | $\widetilde{LAD}^{NW}$ | $\widetilde{LAD}^{M}$ |
|:---:|:---:|:---:|:---:|
| $N \geq 2$ *and* $N < 10$ | 450% | 410% | 416% |
| $N \geq 10$ *and* $N < 15$ | 137% | 234% | 114% |
| $N \geq 15$ *and* $N < 30$ | 99% | 183% | 83% |
| $N \geq 30$ *and* $N < 100$ | 61% | 52% | 51% |
| $N \geq 100$ *and* $N < 1000$ | 37% | 31% | 30% |

## 5. Discussion

The present work extends the method of the theoretically bias-corrected Maximum Likelihood Estimator, initially introduced for the attenuation coefficient [15], to the LAD. The new estimator accounts for vegetation element size, wood volume and hits, correction factors for effective footprint, vegetation clumping and orientation, and multiview data. It can be applied to any volume of interest, for example either a voxel, a crown volume, or even horizontal canopy layers. Our approach can be used as an alternative to Beer's law-based methods in all cases. For example, in a horizontal layer with heights between $h$ and $h + dh$, the gap fraction approach $LAD(h) \approx \frac{-d \log(Pgap)\, cos(\theta)}{Gdh}$ [3] can be replaced by the MLE:

$$LAD(h) \approx \frac{Ni\, cos(\theta)}{G \Sigma h}, \tag{19}$$

where $cos(\theta)$ is the zenith angle, $Ni$ is the number of hits in the, and $\Sigma h$ is the sum of free path heights (which are equal to $dh$ when beams have no interception in the layer, and equals to the difference between the height of hits and $h$ for beams with returns).

As the MLE naturally incorporates variations in view angle and distance to scanner, it should be applicable to UAV LiDAR data, in which beams are emitted from a moving scanner. The application to UAV would require that the traversal algorithm accounts for the UAV travel path and that corresponding correction factors are known. The method also requires estimation of the trajectory of beams with no return, which might be impossible with some lasers. The efficient multiview formulation, as well as bias correction for low beam number, could be especially relevant in the context of UAV.

The novelty of the approach presented here lies in the fact that the Maximum Likelihood Estimation is applied directly to the *LAD* rather than to the attenuation coefficient, as in the previous study [15], and that wood elements are explicitly considered as a volume in which no leaf can be present. This significant advance was permitted by the fact that the MLE does not assume a particular topology for the volume of interest [15], so that it can be applied to a very complex and unknown volume (here, the volume of the voxel which is not occupied by woody elements). On the contrary, Beer's law-based methods cannot be easily applied to an unknown geometry, as shown in Section 4.1, and does not take full advantage of all the information available in free paths [15]. In the present formulation, no assumption is made on the relative distribution of leaf and wood, the only assumption being that leaves are randomly distributed in the volume of the voxel that is not occupied by wood. The random distribution assumption of leaves is not fully realistic, but discrepancies can be corrected through factors to account for leaf orientation, sub-volume clumping [14], and LiDAR effective footprint [14], which were rigorously included in the new approach in a straightforward manner. Although presenting strong similarities with the modified contact frequency first implemented in a previous study [9], the mathematical derivations suggest that beams corresponding to wood hits and those corresponding to non-central leaf hits should be accounted for in the free path sum, contrary to what was suggested

in the previous study [9]. Another difference is the improvement of the manner of accounting for vegetation element size correction suggested in another study [18], which is also different, as already pointed out in another study [15], with the notion of effective free path (Equation (3)). More significant differences should be expected, however, from the difference in free path sum computations than from the difference in element size corrections.

In our formulation, one of the critical aspects is to be able to estimate a fraction of leaf hits $F$, as well as the leaf volume fraction $\alpha$ (Equation (13)). The development of algorithms and methods for leaf and wood separation is a subject of active research [24–29], which is a prerequisite of most methods aiming at retrieving wood volume [22]. One could notice that determining the leaf fraction $F$ is less challenging than the classification of each individual hit as "leaf" and "wood", in the sense that leaf fraction can be correctly estimated from a classification method with significant omission and commission errors inside the voxel. In particular, the leaf fraction can be estimated on a subset of the point cloud, which could help to save computational resources. The correction factor $\alpha$ for wood volumes can probably be neglected in most situations corresponding to foliage, since bulk density of thin twigs are to the order of 0.1 kgm$^{-3}$, which corresponds to volume fraction to the order of 0.02 [30]. However, such a correction is likely to be necessary when trunks or large branches intersect the voxel, otherwise leading to *LAD* overestimation, even if the leaf fraction $F$ is correctly estimated. In this context, tree models derived from LiDAR data [22] can provide the appropriate information.

Our numerical experiments enabled a theoretical validation of the new estimator in two simplified but plausible contexts, as well as a comparison with other former formulations to account for wood returns and to combine multiview data, thanks to well-defined references [2]. These numerical experiments extended the ones of the previous study [15], since the ray tracing and the traversal algorithms were applied within voxels with wood volumes and within a virtual, but more realistic forestry plot, as in previous studies [20,23], rather than within individual voxels. We found that the present formulation was correct in the presence of wood volumes and a large number of wood returns, contrary to previous formulations [9,17]. Also, the multiview estimator performed better than the "*Nmax*" [19] and "*N-weighted*" [8,11] formulations when multiple scans were available, without requiring any additional complexity. Such a result was expected in terms of errors for the "*Nmax*", since this basic approach ignored the information provided by secondary viewpoints. On the contrary, the counter performance of the "*N-weighted*" formulation was relatively unexpected, leading to much higher errors because of infrequent but very large overestimations when one of the poor viewpoints led to an outlier.

This later point highlights the importance of the use of unbiased estimators; more generally, the unbiasedness and efficiency of estimators in the inner-canopy where point density is low is critical [2]. Indeed, our second numerical experiment confirms that the distributions of beam numbers in voxels at various heights is very heterogeneous (Figure 5). Above 6 meters, and up to the top of the canopy, the percentages of unexplored or poorly-explored voxels were very high. Of course, such statistics are highly dependent on the number of scans (here, 5), the scanner angular resolution (here, 0.036°), and the grid size (here, 0.1 m). Such sensitivities, as well as their consequences on estimation accuracy, are analyzed in detail in a previous study [23] and are beyond the scope of the present article, which aimed at presenting the new estimator and some brief validations. It was relevant, however, to recall the frequent occurrence of poorly-explored voxels to highlight the importance of the results of the numerical experiment presented here.

The present study was carried out with MATLAB scripts developed by the authors, as in previous studies [11,14,15,23]. However, the single-view estimator can already be computed in the gridded scene using a plug-in of the COMPUTREE platform (http://computree.onf.fr/?page_id=42) called LVOX (http://computree.onf.fr/?page_id=422) that implements a traversal algorithm, whereas the multiview estimators are currently implemented in LVOX.

## 6. Conclusions

The study confirms the potential of the Maximum Likelihood Estimation method for LAD from single-echo LiDAR data, as already demonstrated in a previous study [15]. The method provides the economy of transmittance computation and inversion that are required in Beer's law-based methods, and is hence more efficient. Our estimator for *LAD* can be used in any volume of interest (voxels, crown volumes, or even thin horizontal layers, as in gap fraction approaches; Equation (19)). A fraction of these volumes can be occupied by wood sub-volumes, and the estimator includes correction factors for vegetation element size, LiDAR effective footprint, leaf orientation, and multiple viewpoints. The only fundamental assumption is that vegetation elements are randomly distributed in sub-volumes that are not occupied by the wood. However, a clumping factor can be used to handle discrepancies due to vegetation morphology and vegetation element clumping in the sub-grid.

The new framework can be applied to any multiview dataset in a straightforward manner, such as multiview TLS. It can probably be extended to UAV LiDAR scanning, provided that a traversal algorithm is available to compute hits and free path distributions, that shooting trajectories are known, and that the different correction factors (vegetation element size, leaf orientation, leaf hit fraction, calibration factors, and wood volume fraction) are available. Beyond its conciseness and mathematical support, our two numerical experiments demonstrated the good performance of the new estimator, which compared favorably to other existing methods. In particular, we showed that several formulations suggested in earlier studies were either incorrect or less efficient. Because it rigorously accounts for all factors that are suspected to affect LAD estimation (with the exception of multi-echoes), we think it should be more widely used and tested in the field against actual references. Ongoing development in the COMPUTREE platform, which is dedicated to LiDAR point cloud processing, should ease the process, whereas the evaluation in field condition is still in progress [23].

**Author Contributions:** conceptualization, methodology, computation, F.P. and M.S.; writing—original draft preparation, F.P.; writing—review and editing, F.P., M.S., and J.-L.D.

**Funding:** This research received no external funding.

**Acknowledgments:** We acknowledge the three anonymous reviewers, which helped to significantly improve the present manuscript in terms of clarity, accuracy, and results.

**Conflicts of Interest:** The authors declare no conflict of interest.

## Appendix A. Estimation of $\lambda_1$ for Simple Vegetation Element Shapes

According to a previous study [15], the attenuation coefficient of a single vegetation element in a cubic voxel of size $\delta$ is:

$$\lambda_1 \approx \frac{S_1}{\delta^3},$$ 
(A1)

where $S_1$ is the cross-sectional area of a single vegetation element.

For a needle of radius $r$ and length $l$, this leads to:

$$\lambda_1 \approx \frac{2\pi rl}{4\delta^3}$$ 
(A2)

For a (small) needle of diameter $2r = 0.5$ mm and length $l = 5$ cm, we have:

$$\lambda_1 \approx 2 \; 10^{-5}\delta^{-3}$$ 
(A3)

For a flat leaf of radius $r$, this leads to:

$$\lambda_1 \approx \frac{2\pi r^2}{4\delta^3}$$ 
(A4)

For a (large) leaf of diameter $2r = 10$ cm, we have:

$$\lambda_1 \approx 5 \ 10^{-3} \delta^{-3} \tag{A5}$$

## Appendix B. Optimized Multiview Estimator in a Voxel of Interest

The following derivation generalized the approach suggested in Section 3 and Appendix C in the previous study [15]. More details on the rationale of the method are provided there.

Here, we assume that we have $M$ scans. We want to compute the Maximum Likelihood Estimator of $LAD$, from $\{N_j\}_{j=1,M}$ beams of different scans. For each scan j, the attenuation coefficient $\lambda_j$ in volume of interest $V - V_w$ corresponds to a projected area of leaf elements equal to $\lambda_j(V - V_w) = c_j LAD \ V$, with $c_j = \frac{G_j}{H_j}$. Hence, $\lambda_j = \frac{c_j LAD}{\alpha}$.

The probability distribution of free path $z$ in the voxel in the context of randomly-distributed elements is:

$$f_j(z;\delta) = \begin{cases} \lambda_j(1 - \lambda_1 z)^{\frac{\lambda_j}{\lambda_1} - 1} & (leaf \ hit) \\ (1 - \lambda_1\delta)^{\frac{\lambda_j}{\lambda_1}} & (no \ leaf \ hit) \end{cases}, \tag{B1}$$

where $S_1$ is the cross-sectional area of a single vegetation element.

Using the effective path $z_e = -\frac{\log(1 - \lambda_1 z)}{\lambda_1}$, (B1) can be rewritten:

$$f_j(z;\delta) = \begin{cases} \lambda_j e^{-(\lambda_j - \lambda_1)z_e} & (leaf \ hit) \\ e^{-\lambda_j z_e} & (no \ leaf \ hit) \end{cases} \tag{B2}$$

Let us denote $\left\{ze_j^i\right\}_{i=1,N_j}$ the $N_j$ "effective" free paths of scan j. From Equation B1, the likelihood of $Z$ is:

$$
\begin{aligned}
\mathcal{L}\left(LAD; \left\{z_j^i\right\}_{i=1,N_j \ \text{and} \ j=1,M}\right) &= \prod_{j=1}^{M}\prod_{i=1}^{N_j} f_j\left(z_j^i; \delta_j^i\right) = \\
&\prod_{j=1}^{M}\prod_{leaf \ hits}\lambda_j e^{-(\lambda_j - \lambda_1)ze_j^i}\prod_{no \ leaf \ hit}e^{-\lambda_j ze_j^i} = \\
&\prod_{j=1}^{M}\left(\lambda_j^{Ni_j^l}\prod_{i=1}^{N_j}e^{-\lambda_j ze_j^i}\prod_{leaf \ hits}e^{\lambda_1 ze_j^i}\right) = \\
&\prod_{j=1}^{M}\left(\left(\frac{LADc_j}{\alpha}\right)^{Ni_j^l}\prod_{i=1}^{N_j}e^{-\frac{LAD}{\alpha}c_j ze_j^i}\prod_{leaf \ hits}e^{\lambda_1 ze_j^i}\right) = \\
&\left(\frac{LAD}{\alpha}\right)^{Ni^l}\prod_{j=1}^{M}\left(c_j^{Ni_j}\left(\prod_{i=1}^{N_j}e^{-c_j ze_j^i}\right)^{\frac{LAD}{\alpha}}\prod_{leaf \ hits}e^{\lambda_1 ze_j^i}\right),
\end{aligned}
\tag{B3}
$$

where $Ni_j^l$ is the number of leaf hit for scan j and $Ni^l = \sum_j Ni_j^l$ is the total number of hits.

The ML estimator is the value $LAD$ that cancels the first derivative of $\mathcal{L}$ [16]. The logarithm of the likelihood is:

$$
\begin{aligned}
&log\mathcal{L}\left(LAD; \left\{z_j^i\right\}_{i=1,N_j \ \text{and} \ j=1,M}\right) \\
&= Ni^l \log\left(\frac{LAD}{\alpha}\right) + \sum_{j=1}^{M}Ni_j^l\log(c_j) - \frac{LAD}{\alpha}\sum_{j=1}^{M}\sum_{j=1}^{N_j}c_j ze_j^i + \sum_{leaf \ hits}\lambda_1 ze_j^i
\end{aligned}
\tag{B4}
$$

Derivation with respect to $LAD$ and equating to zero provides:

$$\frac{dlog\mathcal{L}}{dLAD} = \frac{1}{\alpha}\frac{Ni^l}{\frac{LAD}{\alpha}} - \frac{1}{\alpha}\sum_{j=1}^{M}\sum_{j=1}^{N_j}c_j ze_j^i = 0 \tag{B5}$$

Hence,

$$\text{MLE}_{\text{LAD}} = \alpha \frac{Ni^l}{\sum cz_e}, \tag{B6}$$

with $Ni^l = \sum_j Ni^l_j$ the total number of leaf hits et $\sum cz_e = \sum_{j=1}^{M} \sum_{j=1}^{N_j} c_j ze^i_j$ the sum of product $c_j ze^i_j$ for all beams.

Hence, the ML estimator (also called modified contact frequency) $\frac{1}{c}\widetilde{\lambda} = \frac{I}{cz_e}$ can be generalized to multiple viewpoints.

As explained in the previous study [15], the MLE exhibits a positive bias when the optical path explored within the voxel is limited. Following supplementary C in the previous study [15], we can adapt the bias correction to the multiview formulation. Since $\text{MLE}_{\text{LAD}} = \alpha f(Ni^l, \sum cz_e)$ with $f(x,y) = \frac{x}{y}$, the unbiased estimator $\text{LAD}^m$ can be approximated as:

$$\frac{\widetilde{\text{LAD}}^M}{\alpha} = \frac{Ni^l}{\sum cz_e} - \frac{1}{2}\sigma^2_{Ni^l}\frac{\partial^2 f}{\partial x^2}\left(Ni^l, \sum cz_e\right) - \frac{1}{2}\sigma^2_{\sum cz_e}\frac{\partial^2 f}{\partial y^2}\left(Ni^l, \sum cz_e\right) - \sigma_{Ni^l,\sum cz_e}\frac{\partial^2 f}{\partial x \partial y}\left(Ni^l, \sum cz_e\right) \tag{B7}$$

The different terms can be estimated as follows:

$$-\frac{1}{2}\sigma^2_{Ni^l}\frac{\partial^2 f}{\partial x^2}\left(Ni^l, \sum cz_e\right) = -\frac{1}{2}\sigma^2_{Ni^l} \times 0 = 0 \tag{B8}$$

$$-\frac{1}{2}\sigma^2_{\sum cz_e}\frac{\partial^2 f}{\partial y^2}\left(Ni^l, \sum cz_e\right) = -\sigma^2_{\sum cz}\frac{Ni^l}{(\sum cz_e)^3} \tag{B9}$$

$$-\sigma_{Ni^l,\sum cz_e}\frac{\partial^2 f}{\partial x \partial y}\left(Ni^l, \sum cz_e\right) = \sigma_{Ni,\sum cz_e}\frac{1}{(\sum cz_e)^2} \tag{B10}$$

We now estimate $\sigma^2_{\sum cz_e} = E\left[(\sum cz_e)^2\right] - E[\sum cz_e]^2$ and $\sigma_{Ni^l,\sum cz_e} = E\left[Ni^l \sum cz_e\right] - E\left[Ni^l\right]E[\sum cz_e]$. Because of beam independency, and since $E\left[\overline{z}^2\right] = \frac{2}{\lambda}E\left[\mathbf{1}_{leafhit}z_e\right]$ (Equation (C13) in the previous study [15]) and $\frac{LAD}{\alpha} \approx \frac{Ni^l}{\sum cz_e}$ (Equation (B6)):

$$\begin{aligned}
E\left[(\sum cz_e)^2\right] = &\sum_j c_j^2 E\left[\sum ze_j^2\right] = \sum_j c_j^2 N_j E\left[\overline{ze_j}^2\right] = \sum_j \frac{1}{\lambda_j/1/c_j}2N_j E\left[\mathbf{1}_{leaf\ hit}c_j ze_j\right] \\
&\approx \sum_j \frac{\alpha}{LAD}2\sum_{leaf\ hit} c_j ze_j = \frac{2\alpha}{LAD}\sum_{leaf\ hit} cz_e
\end{aligned} \tag{B11}$$

Similarly:

$$E\left[Ni^l \sum cz_e\right] = \sum_j E\left[\sum \mathbf{1}_{leaf\ hit}c_j ze_j\right] = \sum_{leaf\ hit} cz_e \tag{B12}$$

Hence, the plug-in in Equation (B7):

$$\frac{\widetilde{\text{LAD}}^M}{\alpha} = \frac{Ni^l}{\sum cz_e} - \left(\frac{2\alpha}{LAD}\sum_{leaf\ hit}cz_e - (\sum cz_e)^2\right)\frac{Ni^l}{(\sum cz_e)^3} - \left(\sum_{leaf\ hit}cz_e - Ni^l\sum cz_e\right)\frac{1}{(\sum cz_e)^2} \tag{B13}$$

Hence, because of Equation (B6):

$$\widetilde{\text{LAD}}^M = \frac{\alpha}{\sum cz_e}\left(Ni^l - \frac{\sum_l cz_e}{\sum cz_e}\right) \tag{B14}$$

**Appendix C. A Numerical Experiment to Compare Different MULTIVIEW Formulations**

Method

We conducted a numerical experiment rather than using actual data because attributing error source in actual data is often difficult in this research field [2,13,23]. The goals of this experiment were to (i) provide a theoretical validation of the *"M"* multiview estimator of *LAD* presented above (Equation (15)), and (ii) compare its performance with the two usual formulations to combine single-view estimates ("Nmax" and "N-weighted" $\widetilde{LAD}^{Nmax}$ and $\widetilde{LAD}^{NW}$, Equations (10) and (11)). We first generated a "reference" *LAD* tridimensional field $LAD_{ref}$ in a mesh grid, with voxels of size equal to 0.1 m, corresponding to a cubic vegetation scene with a 10-m lateral extension and a 10-m height. $LAD_{ref}$ corresponded to a clumped spatial distribution simulated from *RandomFields* R package, which was parameterized to correspond to realistic features of natural vegetation. The mean clump size, which was representative of the tree crown diameter, was 4 m, whereas typical *LAD* vertical profiles, as well as a projection function, were implemented. In order to get a more realistic reference field, the random field $LAD_{ref}$ was modified as follows. We multiplied it by a realistic vertical profile to get limited vegetation under 3 m, and a peak in *LAD* around 7 m height (Figure A1a). Also, the first decile of $LAD_{ref}$ values was set equal to 0 in order to generate actual gaps between crowns. Finally, random variations were also introduced to simulate the occurrence of small gaps (~1 m), representative of branch-scale heterogeneity inside tree crowns. These settings led to a clumped vegetation scene with a 70% cover fraction (Figure A1b) and a vertical structure (Figure A1a). The LAI of the virtual scene was about 3.8, which corresponds to a mean $LAD_{ref}$ of 0.38 m-1 (the scene vertical extent was 10 m). Maximal $LAD_{ref}$ values reached 3.8 m$^{-1}$.

A leaf projection function was implemented to complete vegetation properties:

$$G(\theta, z) = \frac{1}{2} + 0.4 \frac{z}{h} \cos(2\theta), \tag{C1}$$

where $\theta$ was the angle between the beams and the vertical, which ranged between 0 and $\pi$.

According to this setting, leaves were planophile near the canopy top $(z \approx h)$, with G = 0.9 for vertical beams $(\theta \approx 0$ or $\theta \approx \pi)$ and 0.1 for horizontal beams $\left(\theta \approx \frac{\pi}{2}\right)$, and random near the ground $(z \approx 0)$, with G = 0.5.

At last, the leaf fraction was parameterized to account for wood and leaf association along the vertical axis following:

$$F(z) = \left(0.1 + 0.8 \frac{z}{h}\right)^2 \tag{C2}$$

The leaf fraction was, hence, equal to 0.9 at canopy top $(z \approx h)$ and 0.1 near the ground $(z \approx 0)$. The vertical profile of $LAD_{ref}$, as well as a two-dimensional horizontal distribution of this vegetation field, are shown in Figure A1a,b. They correspond to a LAI of 3.8 and a cover fraction of 70%.

We then simulated virtual TLS scans processed at five different locations, with a 0.036° angular resolution. Simulations were based on turbid media assumption (assuming that $\lambda_1 \approx 0$, for simplicity), which states that the probability of a beam to be intercepted increases exponentially with the optical depth (product of attenuation coefficient and distance travelled). For simplicity, the volume fraction of wood elements was neglected ($\alpha = 1$). The locations in which individual laser beams were intercepted were, thus, generated from random numbers, as in the previous study [15], but the approach was generalized to a heterogeneous vegetation scene, as in a previous study [23].

The reference attenuation coefficient $\lambda_{ref,j}$ related to $LAD_{ref}$ for a given scan *j* depends on leaf projection, leaf fraction, vegetation heterogeneity, and scanner properties (inverting Equation (1)). Let $\left(x_j, y_j, z_j\right)$ be the coordinates of the scanner corresponding to scan j and $(x, y, z)$ the coordinates of the

center of a voxel in the vegetation scene. The effective attenuation coefficient for both leaf and wood for scan *j* was:

$$\lambda_{ref,j}(x, y, z) = LAD_{ref}(x, y, z) \frac{G_j(x, y, z)}{F(z)H_j(x, y, z)} \tag{C3}$$

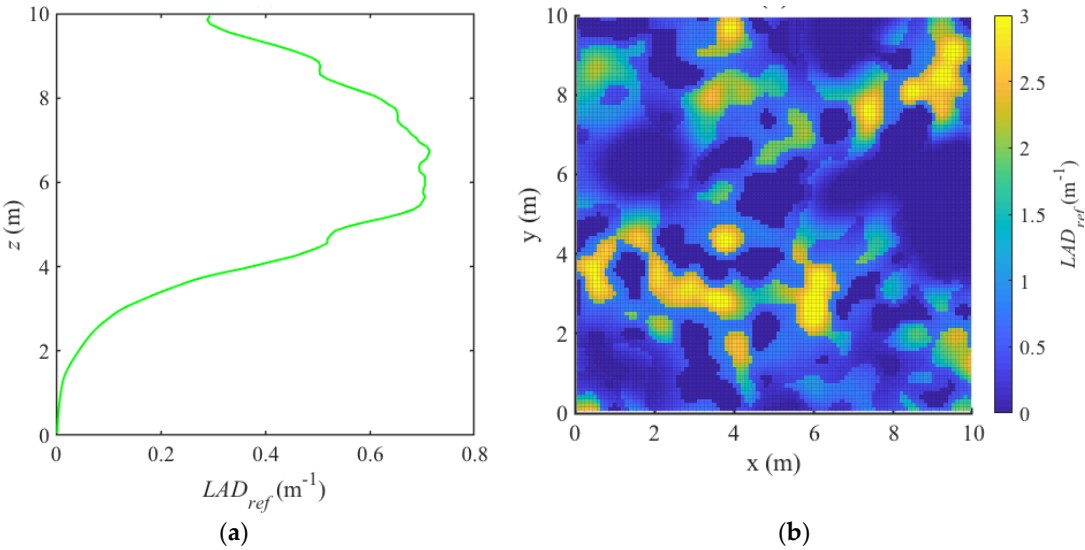

**(a)**                                   **(b)**

**Figure A1.** Reference vegetation: (**a**) vertical profile of $LAD_{ref}$; (**b**) horizontal distribution of $LAD_{ref}$ at $z = 6$ m.

A beam emitted from the scanner j in the direction of $(x, y, z)$ had the following projection function G (since $\cos(2\theta) = \cos(\theta)^2 - \sin(\theta)^2$):

$$G_j(x, y, z) = \frac{1}{2} + 0.4 \frac{z}{h} \frac{(z - z_j)^2 - (x - x_j)^2 - (y - y_j)^2}{(x - x_j)^2 + (y - y_j)^2 + (z - z_j)^2} \tag{C4}$$

We assumed that the distance effect (caused by an increase in effective footprint of the scanner, as identified in a previous study [14]) has the following effect on the attenuation coefficient:

$$H_j(x, y, z) = 1 - 0.05 \sqrt{(x - x_j)^2 + (y - y_j)^2 + (z - z_j)^2}, \tag{C5}$$

which expressed that leaf area was overestimated by a factor of 2 at a distance of 10 m to the scanner ($H_j = 0.5$), which is in agreement with observations in a previous study [14].

We simulated five virtual point clouds corresponding to a scanner located 1 m from the ground and at each corner of the plot and one scan at the center: $(x_1, y_1, z_1) = (7.5, 7.5, 1)$; $(x_2, y_2, z_2) = (7.5, 2.5, 1)$; $(x_3, y_3, z_3) = (2.5, 2.5, 1)$; $(x_4, y_4, z_4) = (2.5, 7.5, 1)$; $(x_5, y_5, z_5) = (5, 5, 1)$. Their shooting patterns corresponded to a 0.036° angular resolution over the horizontal (ranging from 0 to 180°) and the vertical (ranging from 0 to 360°), so that each scan contains around 50 million beams, which is typical of the resolution used in the field [11,14,23]. For each beam, we simulated its eventual hit location with a ray-tracing algorithm. First, the optical path (i.e., initial potential to pass through vegetation) of each beam was randomly simulated according to the Beer's law (assuming infinitely small elements, i.e., $\lambda_1 \approx 0$):

$$l = -\log(p), \tag{C6}$$

with p as a random number within ]0;1], which corresponds to the initial chance to be intercepted by vegetation. We then computed the trajectory of this beam within the computational grid from its

initial position at scanner location by computing the "amount" of the optical path required to cross the next voxel.

This amount was calculated by multiplying the reference attenuation coefficient of this voxel (computed from Equation C3) by the length of the segment corresponding to the intersection of the beam and the voxel. When the residual optical path of the beam was shorter than this amount, a hit occurred within this voxel at a location corresponding to this residual optical path. On the contrary, when the remaining optical path was greater than this amount, it meant that the beam travelled farther than the voxel. The process was recursively applied to the next voxel—the "new" residual optical path corresponding to the remaining of the previous one. The process ended in the case of a hit, or when a beam reached the bounding box of the computational grid. In this later case, the beam was never intercepted in the computational grid, thus corresponding to a beam with no hit. This process was similar to the one used in a previous study [20] to simulate photon trajectories to compute the radiative transfer from a flame through a voxelized heterogeneous vegetation scene with a Monte Carlo approach. Hence, five virtual point clouds were simulated in accordance with $\lambda_{eff,j}$, which accounted for both vegetation and instrument properties.

Finally, we applied a traversal algorithm to each point cloud j to retrieve leaf hits and free path distributions in the voxel (size equal to 0.1 m) in order to compute the different statistics required for the different multiview estimators of the LAD. In particular, the number of hits $N_i$, the number of sampling beams N, and the free path lengths of individual beams were computed in each voxel.

We computed the three multi-#view estimators ($\widetilde{LAD}^{Nmax}$, $\widetilde{LAD}^{NW}$, and $\widetilde{LAD}^{M}$). A two-dimensional horizontal distribution of $\widetilde{LAD}^{M}$ is shown in Figure A2 to illustrate these estimates and can be directly compared to Figure A1b. The blank pixels correspond to locations in which voxels were not sampled by any beam because of vegetation occlusion. The impact of such occlusion was discussed in detail in a previous study [23] and is beyond the scope of the present article.

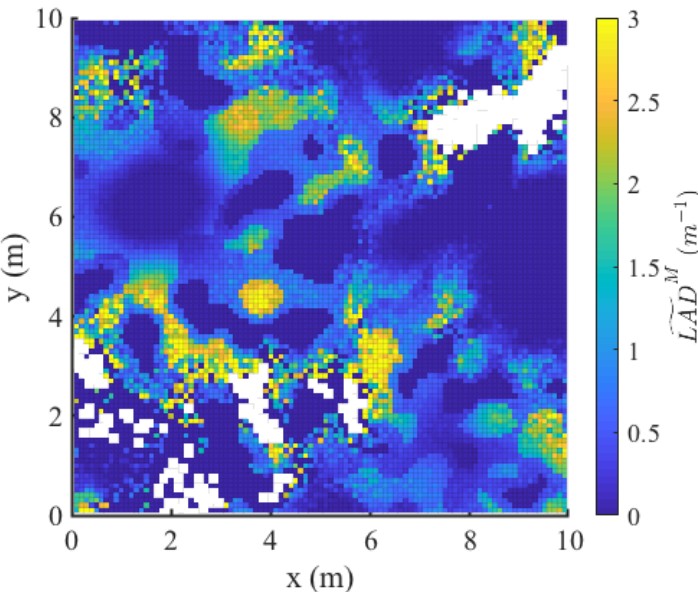

**Figure A2.** Estimated horizontal distribution of $\widetilde{LAD}^{M}$ at z = 6 m. This distribution could directly be compared to $LAD_{ref}$ in Figure A1b. Blank pixels correspond to unexplored voxels, which revealed occluded locations in the canopy.

The performance of the three multiview estimators were compared thanks to reference *LAD* values. We first evaluated their biases by comparing estimated and reference *LAD* values, grouped by classes of total beam numbers exploring voxels (*N*). Indeed, a previous study [15] showed that the magnitude of the biases can strongly vary with the number of sampling beams. Then, we computed

the Root Mean Square Error (RMSE) of the estimations in individual voxels. As for the bias, RMSE was computed per class of total beam numbers exploring voxels (*N*). Both biases and RMSE were expressed in percentage of the mean *LAD* in corresponding voxels in order to ease the interpretation of the results.

Results

Figure A3 shows some comparisons between the three multiview formulation for two classes of beam numbers ($N \in [5, 15[$ and $N \in [100, 500[$). In these examples, Equation (15) leads to the best results, although improvements can be marginal, especially when beam number are larger than 100. Results in the other classes are presented in Tables 2 and 3.

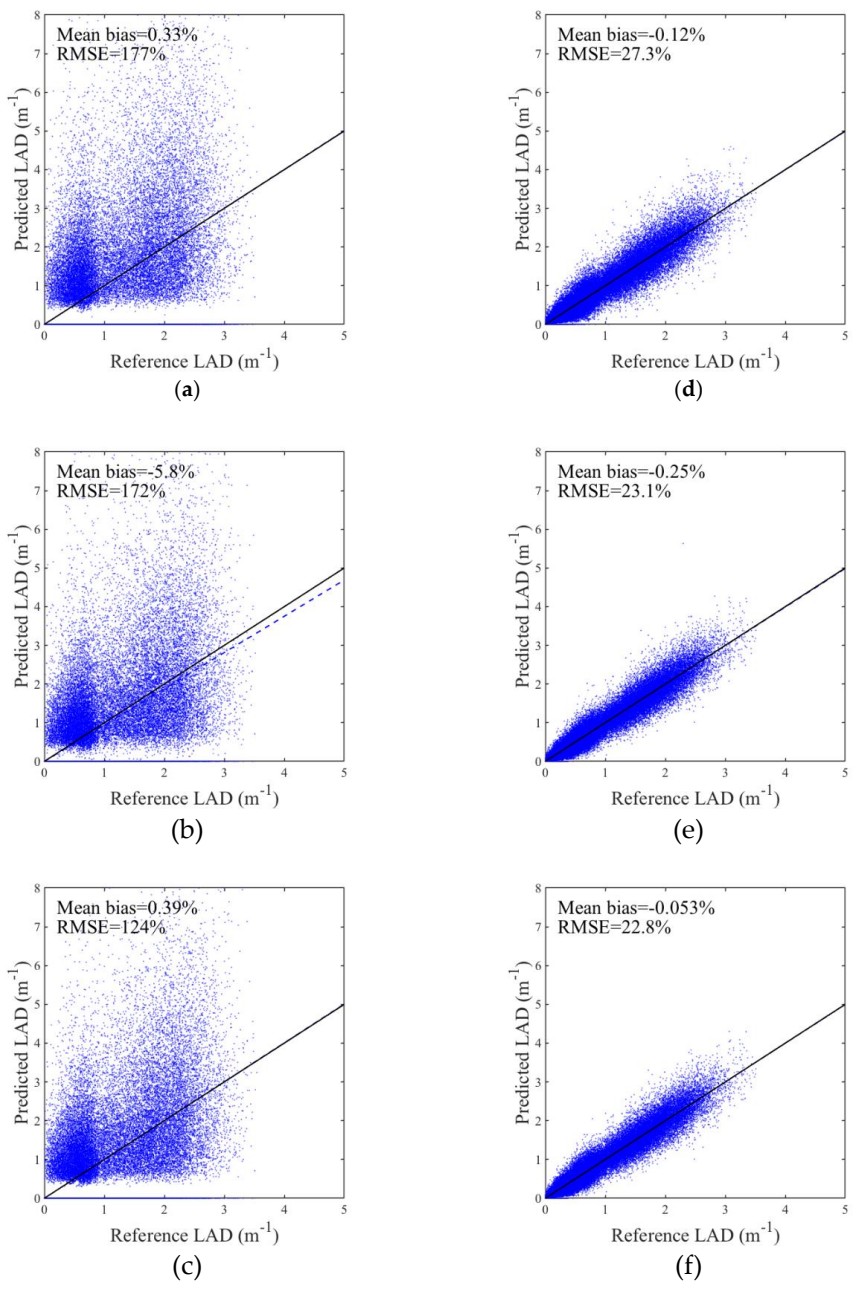

**Figure A3.** Comparison between predicted and reference LAD, for the three multiview formulations for two classes of beam numbers: (**a**) $\widetilde{LAD}^{Nmax}$, $N \in [5, 15[$; (**b**) $\widetilde{LAD}^{NW}$, $N \in [5, 15[$; (**c**) $\widetilde{LAD}^{M}$, $N \in [5, 15[$; (**d**) $\widetilde{LAD}^{Nmax}$, $N \in [100, 500[$; (**e**) $\widetilde{LAD}^{NW}$, $N \in [100, 500[$; (**f**) $\widetilde{LAD}^{M}$, $N \in [100, 500[$.

**Appendix D. Leaf Fraction Corresponding to the 200 Numerical Simulations Presented in Section 4.1**

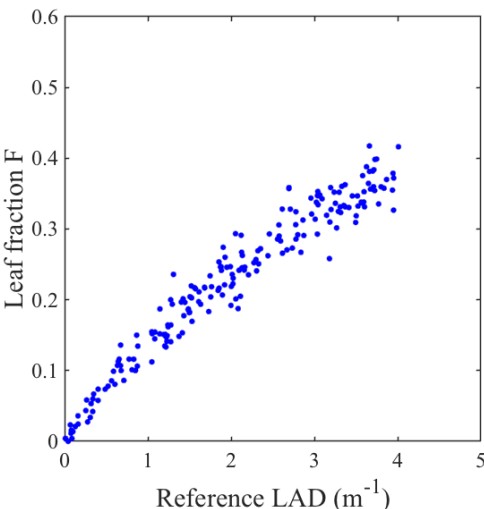

**Figure A4.** Leaf fraction $F = \frac{Ni^l}{Ni}$ in the 200 simulations presented in Section 4.1.

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
