# Peer review of "Accounting for Wood, Foliage Properties, and Laser Effective Footprint in Estimations of Leaf Area Density from Multiview-LiDAR Data"

_remotesensing, doi:10.3390/rs11131580_

Round 1

Reviewer 1 Report

General comments,

The study presents an approach that estimates LAD in 3D space from multiple views of TLS scans. I commend the mathematic rigorousness of the study in pushing forward the application of TLS data to 3D mapping of forest biophysical properties. The current paper needs some changes to clarify the theory and methodology and improve the readability of the manuscripts.

First, the introduction is somewhat misleading. The texts read like the MLE-based approaches are independent of Beer-Lambert’s or Beer’s Law. But clearly from your eq. (1), MLE-based approaches, at least yours, is still fundamentally based on Beer's law. The attenuation coefficient, Lambda, is essentially the notion of optical depth, or in the more common expression of Pgap equation for the radiation regime of plant canopy, the LAI*G/(cosine(zenith_angle)*clumping_index). Therefore, the fundamental essence of the MLE-based method is still Beer's Law. I strongly suggest rephrasing the introduction texts to avoid the idea of disconnection between MLE-based method and Beer's Law, particularly the texts between lines 42 to 43.

Second, please includes some figures and tables of the results into the main texts rather than the appendix to show readers the result data upfront. It helps understand the second half of the Section 3.2 and the section of discussion.

Third, I usually find that for people to understand equations better in the context of application, it is better to include the unit of each term in equations. Not only does it help readers to understand the physical meaning of equation terms, but also it demonstrates that units in your equations check out and the correctness of your equations. List the units of the critical terms here, e.g., H, G, Lambda or lambda, z and z_e.

Detailed comments,

1.      L21 to 23, this sentence is very convoluted and does not make much sense to me, please rewrite it.

2.      L23, “beyond its concision”, is “simplicity” what you meant to say?

3.      L33, “The approach is mostly often based on …”, here it seems like you are implying the MLE-based approaches are the mainstream of estimating LAD from TLS data now. Well, without clarification of the level of details in LAD estimation, this sentence is misleading. We can estimate LAD along canopy height, though not every voxel in 3D space, from the traditional and much more common approaches that base upon Pgap estimation using Beer’s Law and assume canopies are azimuthally similar. Many studies on TLS applications to forest structures described and used this type of approaches. Much more detailed estimates of LAD in 3D space, like what you try to achieve for every voxel, indeed rely upon the MLE-based traversal algorithms. But the texts here around Line 33 do not set up this context. Thus, the trailing sentence is misleading. While the traversal algorithms are emerging for LAD estimation at more detailed spatial levels, as you suggest in the later texts, methods that assume azimuthal similarity in canopies are still common and should not left out of contexts in your texts here. Those methods are what you refer to simply as Beer’s Law based even though MLE-based methods are still based on Beer’s Law. Please rewrite this part of introduction to avoid misleading information.

4.      Section 2.1, It's probably better to introduce the equation (1) at the beginning of this section before explaining how H varies with distance, voxel sizes, and vegetation characteristics, etc. At least that'd be much easier for me to follow the first paragraph of this section.

5.      L84, “effective footprint”, you mean “effective laser footprint”?

6.      L91, what do you mean by vegetation characteristics here? Can you specify? Clumping is one of vegetation characteristics that has been emphasized before.

7.      Line 132, do you mean to say random error decreases with increasing beam number N? "Magnitude of which decreases with increasing beam number N"

8.      Eq. 6, the term N is not introduced or explained.

9.      L138 to 139, do you meant to say, zero hits in a voxel do not necessarily mean no vegetation element at all but only indicate current sampling laser beams have not detected any vegetation element yet.

10.   L147 to 148, Meaning is not clear... difficult to follow. Can you rewrite it?

11.   Eq 11, introduce the term "c" here in eq. 11 since your figure 3 used it. No mention of c whatsoever anywhere in the texts.

12.   L186 to 187, and eq. 11, better to mention how you would get the values of G and H by pointing to the appendix. Otherwise, it is confusing to follow why the equation is able to give you actual estimates of LAD.

13.   Section 3.1, it is better to describe how you would estimate alpha and F here rather than mentioning it later in the discussion. It is confusing to see that you introduce two new parameters to account for wood and foliar materials but no ways to estimate them or obtain their values a priori.

14.   L241, what tool do you use to simulate TLS point cloud from “reference” LAD, put reference papers and brief description here besides the explanation in the appendix.

15.   L275, “subvolume clumping”, the leaf orientation is accounted in your MLE estimator by the G factor. But I do not see anything in your estimator that accounts for leaf clumping!

16.   L327, “randomly distributed inside” inside crown or voxel? if within voxel, your method accounts for clumping, why do you require random distribution of vegetation elements within crown?

Reviewer 2 Report

Review of Accounting for wood, foliage properties and laser effective footprint in estimations of Leaf Area Density from multiview-LiDAR data

The paper seems concerned with improving an existing algorithm to compute Leaf-area density from multi-view Lidar data. The paper is very well written. I have little concern that it will be published quickly as there are only minor issues. I want to see it again before it is accepted though because I think that you did not put enough results.

Main concerns

There are not many references to other papers (23). Five are other papers from the same authors. The goal of the current paper is to improve one of their paper from last year. Among the 18 other papers, most are from two distinct research groups. There is not much variety despite a large body of existing papers on this subject. You put UAV as one of your keyword, but it appears first in the discussion, and there are no references for anything related to UAV. Furthermore, if you are saying that it would work for UAV, wouldn’t citing papers about dense airborne Lidar be pertinent?

Not many results are presented. I understand that you put a lot of meat in the appendix. I’d like to see more figures for the biases in order to understand better what’s happening. For example, with N=20, what is the bias? And for other values of N? And is the bias in relation to N much different for varying levels of wood volume? Appendix C might be better placed in the paper rather than apart. Why do you have figure 4 in the discussion?  It should be in the results. Its caption is not clear enough. What is the vertical profiles of percentages?

Minor concerns

Line 42-45: The sentence is too long. I would put the first part of the sentence after the second part as it relates more to the next sentence.

Line 48-50: Two ‘simply’ in the same sentence.

Line 97: is that equation used? Its not clear from the context.

There should be more implications of your results in the conclusion.

What software did you use? Is there a package to run your algorithm? Do you plan on releasing it?

Reviewer 3 Report

The introduction is particularly concise and precise.

The manuscript start with the presentation of the previous work of [7] and [8]. But the reading of [8] is required to fully understand the present work.

It is mainly the presentation of the new formulation taking into account multiple views and the wood part assuming a perfect knowledge of the previous works.

Figure 1 presents the MLE method.
Figure 2 presents the notion of wood volume

Figure 3 is an extension of the MLE to multiple views removing the detection of point on wood

The authors insist, in the introduction like all along the text, on the difference between the MLE attenuation coefficient method and other transmittance or the gap fraction Beer’s law-based methods.

Could we have a figure showing the main conceptual difference of the Beer’s law-based methods?
How can you be certain that you don’t have any multiple laser beam reflections? I have to look at [8] to find a discussion about multiple echoes which could also change a bit the results. I also find a reference about the fact that you need retrieving beam trajectory even when no return was registered. This seems quite difficult to transpose to UAV methods.

Only Appendix C evaluates the impact of the Multiview approach. But Table C2 doesn’t show real evidence of improvements.

A discussion about the effectiveness of the distinction between wood and leaves detection is also missing. What happen when trunk are covered by climbing plant or mosses?

This work being a development of [8] already published it can be also published for the formalism of the new developments. Real evaluation such as Figure 5 in [8] are however missing here and not fully replaced by the Appendix C.

Multiple view are common between parallel swaths of airborne LiDAR (including UAV). A discussion about it could help to understand what could we gain with this new approach, particularly in a remote sensing journal.

Round 2

Reviewer 1 Report

The authors have addressed all the concerns I had in the previous version. A few minor comments,

1.     P2, L51-52, “This trend can probably be explained by the strong legacy of gap fraction approaches in this research field.” Well, only partly true, not as if people dislike changes and cling to legacy, but gap-fraction based methods can use very cheap digital hemispherical photographs as data to estimate LAI and even LAI vertical profiles by raising up cameras vertically. The costs of TLS and the fact that TLS applications to forests is relatively new compared to other data collection approaches (DHP, LiCor LAI-2000 etc.), both are strong factors for gap-fraction methods to remain popular. Please rephrase this statement.

2.     Table 1 and Figure 4, the six methods in the comparison have confusing labels. Please make them distinct with each other. In Table 1, maybe Eq. 8 versus Eq. + alpha factor? In Figure 4 caption, (a) and (d) have the same label, (b) and (e) have the same label  

3.     P9, L 313-314, “This finding is all the more important as … ” It reads unclear … Please rewrite.

4.     P13, L 432 – 433, the references to the leaf-wood separation are lacking some recent representative papers that utilizes spectral and/or spatial information of TLS point clouds.

Ma, L., Zheng, G., Eitel, J.U.H., Moskal, L.M., He, W., Huang, H., 2016. Improved Salient Feature-Based Approach for Automatically Separating Photosynthetic and Nonphotosynthetic Components Within Terrestrial Lidar Point Cloud Data of Forest Canopies. IEEE Trans. Geosci. Remote Sens. 54, 679–696. https://doi.org/10.1109/TGRS.2015.2459716

Li, Z., Schaefer, M., Strahler, A., Schaaf, C., Jupp, D.L.B., 2018. On the utilization of novel spectral laser scanning for three-dimensional classification of vegetation elements. Interface Focus 8, 20170039. https://doi.org/10.1098/rsfs.2017.0039

Vicari, M.B., Disney, M., Wilkes, P., Burt, A., Calders, K., Woodgate, W., 2019. Leaf and wood classification framework for terrestrial LiDAR point clouds. Methods Ecol. Evol. 10, 680–694. https://doi.org/10.1111/2041-210X.13144

Author Response

Here are our detailed answers to the last four concerns:

We added "provided by LiDAR" ln 49 to remind that our text was written in the context of LiDAR and added ", which was focused on 2D sensors such as hemispherical photographs or LiCors, prior to the emergence of more expensive and more complex 3D sensors" at the end of the problematic sentence.

We added ", with alpha multiplicative factor" where was missing. Thank you for catching the missing labels.

The reviewer is correct, this sentence was unclear. We modified as follows: "Another limitation of these last two estimators is that their biases vary with the location of wood volumes inside the voxel, contrary to Eq. 15, which is insensitive to wood volume distributions (provided that leaves are randomly distributed outside these volumes, as in our simulation). For example, the mean bias presented in Figure 4d reached 53% when the branch was located near the trailing face of the voxel (where beams leave the voxel), whereas it was limited to 8% when the branch was located near the leading face of the voxel (where beams enter the voxel), instead of 32% when the branch was centered (as in Figure 4d)."

We added the four suggested references.

Reviewer 2 Report

Good!

Author Response

No comment to answer